# PTC596-Induced BMI-1 Inhibition Fights Neuroblastoma Multidrug Resistance by Inducing Ferroptosis

**DOI:** 10.3390/antiox13010003

**Published:** 2023-12-19

**Authors:** Giulia Elda Valenti, Antonella Roveri, Rina Venerando, Paola Menichini, Paola Monti, Bruno Tasso, Nicola Traverso, Cinzia Domenicotti, Barbara Marengo

**Affiliations:** 1Department of Experimental Medicine, General Pathology Section, University of Genoa, 16132 Genoa, Italy; giuliaelda.valenti@edu.unige.it (G.E.V.); nicola.traverso@unige.it (N.T.); barbara.marengo@unige.it (B.M.); 2Department of Molecular Medicine, University of Padua, 35128 Padua, Italy; antonella.roveri@unipd.it (A.R.); rina.venerando@unipd.it (R.V.); 3Mutagenesis and Cancer Prevention Unit, IRCCS Ospedale Policlinico San Martino, 16132 Genoa, Italy; paola.menichini@hsanmartino.it (P.M.); paola.monti@hsanmartino.it (P.M.); 4Department of Pharmacy, University of Genoa, 16148 Genoa, Italy; bruno.tasso@unige.it

**Keywords:** neuroblastoma, multidrug resistance, BMI-1, P53, glutathione, ferroptosis

## Abstract

Neuroblastoma (NB) is a paediatric cancer with noteworthy heterogeneity ranging from spontaneous regression to high-risk forms that are characterised by cancer relapse and the acquisition of drug resistance. The most-used anticancer drugs exert their cytotoxic effect by inducing oxidative stress, and long-term therapy has been demonstrated to cause chemoresistance by enhancing the antioxidant response of NB cells. Taking advantage of an in vitro model of multidrug-resistant (MDR) NB cells, characterised by high levels of glutathione (GSH), the overexpression of the oncoprotein BMI-1, and the presence of a mutant P53 protein, we investigated a new potential strategy to fight chemoresistance. Our results show that PTC596, an inhibitor of BMI-1, exerted a high cytotoxic effect on MDR NB cells, while PRIMA-1^MET^, a compound able to reactivate mutant P53, had no effect on the viability of MDR cells. Furthermore, both PTC596 and PRIMA-1^MET^ markedly reduced the expression of epithelial–mesenchymal transition proteins and limited the clonogenic potential and the cancer stemness of MDR cells. Of particular interest is the observation that PTC596, alone or in combination with PRIMA-1^MET^ and etoposide, significantly reduced GSH levels, increased peroxide production, stimulated lipid peroxidation, and induced ferroptosis. Therefore, these findings suggest that PTC596, by inhibiting BMI-1 and triggering ferroptosis, could be a promising approach to fight chemoresistance.

## 1. Introduction

Neuroblastoma is an extracranial solid tumour originating from the neural crest, and it is responsible for 15% of all paediatric cancer deaths. More than 50% of newly diagnosed patients are stratified into the high-risk group and are treated with the standard regimens including induction, consolidation, and maintenance therapy [1,2]. Although myeloablative therapy and immunotherapy [3] lead to an improvement in overall survival, a large percentage of high-risk patients are characterised by minimal residual disease that is refractory to induction chemotherapy and induces relapse even after consolidation therapy. In fact, the induction regimen is the most critical step in the treatment of these patients, since resistant clones can emerge during this phase. Therefore, in order to improve the cure rate, it is fundamental to counteract the onset of chemoresistance by killing the greatest number of malignant cells during induction therapy. In fact, it has been widely demonstrated that cancer cells, during long-term therapy, adapt well to drug-induced stress conditions and become refractory to the cytotoxic action of the treatment.

Chemoresistance is mainly due to mutations in genes coding for proteins involved in cell death and survival and to epigenetic changes. Since the most-used anticancer drugs in induction therapy (i.e., etoposide and doxorubicin) exert their cytotoxic action by inducing oxidative stress, therefore producing potentially genotoxic free radicals, mutations in the genes encoding for proteins involved in the modulation of the cellular redox state are frequently found. All these mutations make cancer cells able to escape apoptosis and adapt to oxidative stress [4,5]. Indeed, the enhanced levels of antioxidant molecules such as GSH and the activation of GSH-dependent enzymes have been widely reported to be involved in the adaptive response of cancer cells to therapy [6,7,8].

In this context, alterations of P53, the most studied and frequently mutated gene in human cancers, are associated with adverse clinical prognosis [9]. Taking into account the physiological role of P53 and considering the consequences that its mutations could have on carcinogenesis and therapy resistance, the P53 protein can be considered an excellent target for anticancer therapies. Therefore, several strategies have been proposed to restore P53 function and, among the compounds developed for the reactivation of mutated P53 [10], PRIMA-1^MET^ (2-(hydroxymethyl)-2-(methoxymethyl)quinuclidin-3-one or APR-246) has been shown to boost antitumor activity in several cancer cells [11,12,13,14]. PRIMA-1^MET^ is the first compound capable of reactivating mutant P53 that has reached the clinical trial stage, either as a single drug [15,16] or in combination with anticancer agents [17]. Moreover, focusing the attention on the mechanism of action, it has been found that PRIMA-1^MET^, through the depletion of GSH, increases the production of reactive oxygen species (ROS), which, at the mitochondrial level, induce lipid peroxidation and trigger cell death [18,19,20].

In addition, P53 can be modulated by BMI-1, an oncogene with several functions, including the ability to regulate GSH production [21]. Notably, on the one hand, BMI-1 is able to directly interact and stabilize P53, showing its role in the inhibition of cellular proliferation, and on the other, it can also negatively regulate P53 expression by promoting its ubiquitination and degradation, supporting tumour development [22]. Therefore, considering all the functions of BMI-1, it is reasonable to believe that its dysregulation can be a crucial factor in the onset of cancer [23]. Indeed, the overexpression of BMI-1 is associated with cancer stemness, epithelial–mesenchymal transition (EMT) induction, chemoresistance and, thus, invasion, metastasis formation, and poor prognosis [24,25,26,27,28,29,30,31,32]. Then, BMI-1 inhibition can be a promising approach to fight cancer [33,34,35,36,37]. In this regard, it has been reported that PTC596, a compound able to reduce BMI-1 content, counteracted cancer development by inducing P53-independent apoptosis [38,39] and completely eradicated multiple myeloma in an in vivo model [40].

In this context, our previous studies demonstrated that an etoposide-resistant human neuroblastoma (NB) cell line displays high levels of GSH and shows a mono-allelic deletion of the *13q14.3* locus that leads to miRNA 15a/16-1 downregulation and the consequent overexpression of BMI-1 [28]. Interestingly, etoposide-resistant NB cells (HTLA-ER) and the parental cells (HTLA-230) have been demonstrated to have the same homozygous TP53 missense P53 mutation, p.A161T, and no changes in the expression of MDM2, the endogenous inhibitor of P53 [28].

Taking into consideration the wide literature supporting the role of GSH in cancer progression and therapy refractoriness [8], the depletion of its intracellular levels can be a valid approach to circumvent multidrug resistance. With regard to the possibility of limiting the availability of GSH, different strategies have been employed, such as the inhibition of key enzymes or precursors of GSH synthesis, the consumption of its intracellular stores, and the promotion of its efflux [41]. Among the inhibitors of GSH biosynthesis, buthionine sulfoximine (BSO) has restricted clinical use due to its short half-life and its non-selective effect on healthy and cancer cells [42]. In order to bypass these issues, new strategies are needed to circumvent multidrug resistance. Therefore, taking advantage of our experimental model that is able to mimic in vivo conditions and the P53 and BMI-1 connection, the aim of the present study was to investigate whether the use of PRIMA-1^MET^ and/or PTC596 could be a new and effective approach to counteract the drug resistance of NB cells by reducing intracellular GSH levels.

## 2. Materials and Methods

### 2.1. Chemicals

Etoposide was purchased from Calbiochem (Merk KGaA, Darmstadt, Germany), PRIMA-1^MET^ from Abcam (Cambridge, UK), and PTC596 from PTC Therapeutics (South Plainfield, NJ, USA). Doxorubicin, cyclophosphamide, cisplatin, vincristine, and carboplatin were obtained from Selleck Chemicals LLC (Houston, TX, USA). Ferrostatin-1, liproxstatin-1, RSL3, and erastin were acquired from MedChemExpress (Monmouth Junction, NJ, USA). Stock solutions of these compounds were prepared using sterile water or dimethyl sulfoxide (DMSO, Sigma-Aldrich, St. Louis, MO, USA) as a solvent.

### 2.2. Cell Cultures

HTLA-230, a human MYCN-amplified NB cell line at stage IV, was a gift from Dr. L. Raffaghello (Istituto G. Gaslini, Genoa, Italy), while the etoposide-resistant cell line (HTLA-ER) was selected in our laboratory, as previously reported [28,43]. Cells were periodically tested for mycoplasma contamination (Mycoplasma Reagent Set, Aurogene s.p.a, Pero, PV, Italy). Cells were cultured in RPMI-1640 (Euroclone s.p.a., Siziano, PV, Italy) and supplemented with 10% foetal bovine serum (FBS; Euroclone s.p.a.), 2 mM glutamine (Euroclone s.p.a.), 1% penicillin/streptomycin (Euroclone s.p.a.), 1% sodium pyruvate (Sigma-Aldrich), and a 1% amino acid solution (Sigma-Aldrich).

### 2.3. Treatments

HTLA-230 and HTLA-ER were treated for 48 h with 0.625–125 µM of etoposide, 0.025–5 µM of doxorubicin, 0.05–10 mM of cyclophosphamide, 0.165–33 µM of cisplatin, 5–1000 µM of carboplatin, and 1.25–250 nM of vincristine. In other experiments, both cell populations were exposed for 48 h to the compounds given alone or in combination (as in the induction therapy) at a concentration comparable to that clinically used [44,45,46]. In order to identify the adequate concentrations to carry out the experiments with PRIMA-1^MET^ and PTC596, pilot studies were performed by treating HTLA-230 and HTLA-ER cells with 10–60 µM of PRIMA-1^MET^ or with 20–200 nM of PTC596 for 48 or 72 h. Moreover, in another series of experiments, HTLA-230 and HTLA-ER cells were treated with ferrostatin or liprostatin as ferroptosis inhibitors, and with RSL3 or erastin as ferroptosis inducers, at doses of 2.5 µM, 1 µM, 50 nM, and 2.5 µM, respectively, for 48 h or 72 h, alone or in combination with 35 nM of PTC596. Cell cultures were carefully monitored before and during the experiments to ensure that the cell density did not exceed 90% of confluence. In order to exclude the interference of DMSO, employed to solubilize these compounds, cells were exposed to the highest DMSO concentration used, and the pilot studies were carried out in parallel with all performed analyses.

### 2.4. Cell Viability Assay (MTS)

In order to assess cell viability/proliferation, the 3-(4,5-dimethylthiazol-2-yl)-5-(3-carboxymethoxyphenyl)-2-(4-sulfophenyl)-2H-tetrazolium (MTS) CellTiter 96^®^ AQ_ueous_ One Solution cell proliferation assay (Promega, Madison, WI, USA) was used according to the manufacturer’s instructions. Briefly, cells (10^5^ cells/well) were seeded in 96-well plates (Corning Incorporated, Corning, NY, USA) and treated with the above-cited drugs, alone or in combination. Next, cells were incubated with CellTiter under standard conditions (37 °C in a humidified incubator with 5% CO_2_). The conversion of tetrazolium to tetrazane takes place only in living cells, and the amount of intracellular formazan can be measured spectrophotometrically. Absorbance was recorded at 490 nm using a microplate reader (EL-808, BIO-TEK Instruments Inc., Winooski, VT, USA). The IC_50_ was evaluated by using GraphPad Prism 8.4.2 software (GraphPad Software, Boston, MA, USA).

### 2.5. Total Protein Extraction and Quantification

At the end of the treatments, to extract total proteins, cells were detached, collected, and centrifuged at 117× *g* for 8 min at 8 °C. Then, the pellet was resuspended in lysis buffer (50 mM Tris HCl, 150 mM NaCl, 2 mM EDTA, 1 mM EGTA, 50 mM EGF, and 1 mM PMSF) containing a mixture of 7X protease inhibitors (Complete Mini Protease, Roche, Zurich, Switzerland) and 1% Triton X-100 (Sigma-Aldrich). The suspension was further lysed by passing it 10 times through a 25-gauge needle-equipped syringe and then centrifuged at 15,000× *g* for 10 min at 8 °C (Heraeus Centrifuge Biofuge 28RS). The supernatant was collected, and its protein concentration was determined with the PierceTM BCA Protein Assay Kit (Thermo Fisher Scientific, Waltham, MA, USA). The colour intensity of the adduct was determined via spectrophotometer analysis at a 570 nm wavelength (Bio-Rad iMark plate reader, Bio-Rad, Hercules, CA, USA). Total protein amounts of the samples were calculated by referring to a calibration curve obtained using bovine serum albumin (BSA) at various concentrations (2–20 mg/mL).

### 2.6. Western Blot Analysis

Protein samples (35 µg) were mixed with 3.5x loading dye (62.5 mM Tris HCl (pH 6.8), 2% SDS, 25% glycerol, 0.01% bromophenol blue, and 62.5 mM β-mercapto-ethanol), denatured at 100 °C × 5 min, and separated via electrophoresis, using 4–20% Mini-PROTEAN TGX ™ Precast Gels (Bio-Rad) along with MW protein markers (Sharpmass VII, Euroclone s.p.a.). At the end of the electrophoretic run, the proteins were transferred via electroblotting onto a Hybond P 0.45 PVDF membrane (Polyvinylidene fluoride, GE Healthcare Amersham, Amersham, England), which was subsequently stained with a solution of Red Ponceau. The non-specific binding sites were blocked by incubating membranes with 5% dry non-fat milk in PBS–Tween (80 mM Na_2_PO_4_, 20 mM NaH_2_PO_4_, 100 mM NaCl, and 0.1% Tween 20) for 1 h at room temperature. Then, the membranes were incubated overnight with rabbit antibody anti-BMI-1, anti-P53, anti-BAX, anti-Bcl-2, anti-N-cadherin, anti-β-catenin, and anti-SNAIL antibodies (Cell Signaling Technology Inc., Danvers, MA, USA, Upstate, Lake Placid, NY, USA), anti-p21 antibodies (Santa Cruz Biotechnology, Inc., Santa Cruz, CA, USA), and mouse antibody anti-MDM2 (Santa Cruz Biotechnology). Then, the membranes were washed in TBS–Tween (20 mM Trizma base, 0.5 M NaCl, and 0.1% Tween 20) and further incubated for 1 h with the rabbit or mouse HRP-conjugated secondary antibody (Cell Signalling). The signal was detected using an enhanced chemiluminescence system detection kit (Pierce™ ECL Western Blotting Substrate, Thermo Fisher Scientific) and analysed with an image densitometer connected to Quantity One software version 4.2.0 (Bio-Rad Laboratories, Hercules, CA, USA).

### 2.7. Clonogenic Assay

HTLA-230 and HTLA-ER cells (150 per well) were seeded in 6-well plates (Corning) and treated with the drugs. Then, the medium was changed, and the cells were maintained in a drug-free medium for 20 days. Finally, the cells were fixed with methanol and stained with crystal violet solution (0.5% in water with 50% methanol). Then, colonies consisting of 30 or more cells were counted under a microscope [47].

### 2.8. Cancer Stem Cell Formation Assessment

To assess cancer stem cell (CSC) formation, floating cells were harvested from 2D cultures, centrifuged (117 rcf for 8 min), and cultured in DMEM-F12 Knock-out (Life Technologies, Carlsbad, CA, USA) containing 1% penicillin/streptomycin (Euroclone s.p.a.), 2% B27 (Life Technologies), 40 ng/mL of basal fibroblast growth factor (bFGF) (R&D Systems, Inc., Minneapolis, MN, USA), and 20 ng/mL of epidermal growth factor (EGF) (Life Technologies) according to [48]. CSCs were disaggregated once a week to promote propagation and resuspended in 50% fresh medium and 50% of the medium in which the cells were grown [48]. In order to evaluate CSC propagation, at any split, CSCs were disaggregated and counted by using a Burker chamber.

### 2.9. H_2_O_2_ Level Determination

HTLA-230 and HTLA-ER cells (10^4^ cells/well) were seeded in 96-well plates (Corning) and treated. Then, cells were stained with 2′-7′ dichlorofluorescein-diacetate (DCFH-DA; Sigma-Aldrich) and incubated with 90% DMSO for an additional 10 min in the dark [43,49]. The generated fluorescence intensity was monitored with a Perkin Elmer fluorometer (Perkin Elmer Life and Analytical Sciences, Shelton, WA, USA) at 485/530 nm excitation/emission. Values were normalised to the protein content.

### 2.10. HPLC Analysis of Intracellular GSH Levels

The intracellular levels of total GSH and oxidised GSH (GSSG) were assessed with high-performance liquid chromatography (HPLC), according to the method reported by Reed for GSH [50] and by Asensi for GSSG [48,51,52]. At the end of the treatments, for the analyses of GSH, cells were collected in a solution composed of 70% perchloric acid, 15 mM bathophenanthrolinedisulfonic acid disodium salt (PBDS), and PBS, while for the analysis of GSSG, 15 mM N-ethyl-maleimide (NEM) was added to this solution to block the free thiol groups. Then, the samples were centrifuged at 15,000× *g* for 15 min, and the supernatant was collected. For the analysis of total GSH, the free thiol groups were blocked with iodacetic acid (at an alkaline pH of 8–9, for 10 min, at room temperature, and in the dark). The pellet was used for protein assay. Finally, 1% 1-fluoro-2,4-dinitrobenzene was added to each sample as a derivatising agent and left in the dark at 4 °C overnight. Then, for quantitative determination, derivatised analytes were injected into HPLC equipped with a –NH_2_ Spherisorb column and a UV detector set at 360 nm, with a flow rate of 1.5 mL/min, with gradient elution. The mobile phase was maintained at 80% solution A (80% methanol in water) and 20% solution B (0.5 M sodium acetate in 64% methanol in water) for 10 min, followed by a 10 min linear gradient to 1% A and 99% B. In order to calculate the GSH and GSSG contents, the areas of the peaks of interest were compared with the areas obtained from external standards, i.e., commercial GSH and GSSG, at known amounts and treated with the same procedure as the samples. The obtained results were normalised with the total amount of proteins in each sample and expressed as µM/µg protein.

### 2.11. Lipid Peroxidation Assay

Lipid peroxidation was measured using the Image-iT™ Lipid Peroxidation Kit (Thermo Fisher Scientific), which allows the detection of lipid peroxidation in live cells via the oxidation of BODIPY™ 581/591 C11 reagent. Specifically, cells were plated in 96 wells and treated as reported above. After treatments, cells were incubated with BODIPY reagent for 30 min at 37 °C as indicated by the supplier. Then, the reagent was removed, and the cells were washed with PBS. Fluorescence was monitored with a Perkin Elmer fluorometer (Perkin Elmer Life and Analytical Sciences, Shelton, WA, USA) first at 580/590 nm excitation/emission and then at 485/520 nm, and the ratio was used for data analysis.

### 2.12. GPX4 Activity

GPX4 activity was evaluated using standard methods [53]. In detail, cell pellets were resuspended in lysis buffer (0.1 M Tris-HCl, 0.25 M sucrose, and protease inhibitors, at pH 7.5), sonicated, and then used in the test (0.1–0.2 mL of the sample per test). Samples were mixed with the assay buffer (0.1 M Tris-HCl (pH 7.8), 5 mM EDTA, 5 mM GSH, 0.1% (*v*/*v*) Triton X-100, 0.16 mM NADPH, and 0.6 IU/mL glutathione reductase (GR)) and incubated for 5 min at 25 °C. Then, the baseline was recorded at 340 nm for approximately 1 min, and finally, the enzymatic activity was started by adding phosphatidylcholine hydroperoxide (0.020 mM). The quantification of the activity was performed on the basis of the net speed with which the absorbance decreased after the addition of the substrate (net speed = speed after the addition of the substrate—baseline speed).

### 2.13. Statistical Analysis

The results are expressed as means ± SEM of at least four independent experiments. The statistical significance of the parametric differences between the experimental data sets was assessed using one-way ANOVA and Dunnett’s test for multiple comparisons. *p* < 0.05 was considered statistically significant.

## 3. Results

### 3.1. HTLA-ER Developed a Multidrug-Resistant Phenotype

Our previous study demonstrated that chronic exposure to etoposide was able to select a drug-resistant NB cell population (HTLA-ER), which was found to also be resistant to doxorubicin [43]. Based on this evidence, it is conceivable that HTLA-ER could have acquired a multidrug-resistant (MDR) phenotype, becoming resistant to several chemotherapeutic drugs [5,54].

Therefore, to this aim, both parental HTLA-230 and HTLA-ER cells were treated with different anticancer drugs (e.g., etoposide, doxorubicin, cyclophosphamide, cisplatin, carboplatin, and vincristine) included in the NB induction therapy [55]. The drugs were administered for 48 h at increasing concentrations, including those mimicking the clinically used doses [44,45,46]. As shown in Figure 1, the treatment with all these drugs reduced the viability of HTLA-230 cells in a concentration-dependent manner, while it determined a slight reduction in HTLA-ER cell viability, only at doses higher than those clinically used (Figure 1a–f).

Indeed, as reported in Table 1, the IC_50_ values of all drugs tested on HTLA-ER were higher (by at least three-fold) in comparison with those evaluated in HTLA-230 cells. Moreover, as shown in Figure 1g, the drug mixture, mimicking the induction therapy, reduced the viability of parental cells by 45%, while it was totally ineffective on HTLA-ER. These data confirm the hypothesis that HTLA-ER cells acquired an MDR phenotype.

### 3.2. PTC596 Affects the Viability of Parental and HTLA-ER Cells and Reduces BMI-1 Expression Levels

In a recent study, we demonstrated that the acquisition of drug resistance by HTLA-ER cells is independent of the TP53 mutational status, and it is associated with a mono-allelic deletion in the *13q14.3* locus, where the miRNA 15a/16-1 cluster is located. Notably, it has been also reported that BMI-1, a direct target of miRNA 15a/16-1, is overexpressed [28,56].

Based on these findings, both NB cell populations were treated with PRIMA-1^MET^, an activator of the mutant P53 protein [11,12,13,14] that has been found to facilitate apoptosis induction in cancer cells [57], and with PTC596, a BMI-1 inhibitor currently undergoing clinical trials [38,58]. In order to identify the adequate concentrations to carry out the experiments with PRIMA-1^MET^ and PTC596, preliminary studies were performed. The doses of both compounds were selected to induce a 25% reduction in cell viability, avoiding exerting a too-marked cytotoxic effect that could have masked etoposide sensitization. Therefore, as shown in Figure 2, 60 µM of PRIMA-1^MET^ (Figure 2a) and 35 nM of PTC596 (Figure 2b) were identified to be the useful doses.

Then, both NB cell populations, prior to incubation with etoposide, were pre-treated with PRIMA-1^MET^ (60 μM) and/or PTC596 (35 nM) for 48 h and 72 h. As shown in Figure 3, 24 h treatment with 1.25 μM of etoposide decreased the viability of HTLA-230 cells by 10%, whereas, as expected, it did not change the viability of HTLA-ER cells compared with untreated ones. Incubation with 60 μM of PRIMA-1^MET^ alone for 48 h and 72 h reduced the viability of HTLA-230 cells by 25% (Figure 3a), while the same treatment did not significantly modify the viability of HTLA-ER cells (Figure 3b). Meanwhile, 35 nM of PTC596 per se affected the viability of both cell populations by 26% and 36% after 48 h and 72 h in comparison with untreated ones, respectively (Figure 3b). Furthermore, pre-incubation with PRIMA-1^MET^ further reduced the viability of etoposide-treated HTLA-230 cells by 18%, while it did not alter HTLA-ER cell viability (Figure 3b). Noteworthily, the pre-treatment with PTC596 markedly decreased the viability of both etoposide-treated NB cells by approximately 30–35% with respect to etoposide given alone.

Furthermore, the combined treatment with PTC596 (72 h) and PRIMA-1^MET^ (48 h) reduced the viability of HTLA-230 cells by approximately 45% (Figure 3a) and that of HTLA-ER cells by 37% (Figure 3b) in comparison with the respective untreated cells. Moreover, this drug combination was more effective than PRIMA-1^MET^ alone, leading to a further 28% and 35% decrease in the viability of parental and resistant cells, respectively, while it was able to reduce only the viability of parental cells by a further 18% with respect to PTC596 alone (Figure 3). The combined treatment with PTC596 (72 h), PRIMA-1^MET^ (48 h), and etoposide (24 h) caused a decrease in the HTLA-230 and HTLA-ER cell viabilities by 42% and 55%, respectively, with respect to PRIMA-1^MET^ + etoposide, by 34% and 10% in comparison with PTC596 + etoposide, and by 20% and 9% compared with PTC596 + PRIMA-1^MET^ (Figure 3).

Therefore, taking into consideration all the results obtained, the subsequent experiments were performed by treating both NB cell populations with 1.25 μM of etoposide (24 h), 60 μM of PRIMA-1^MET^ (48 h), and 35 nM of PTC596 (72 h) given alone or in combination (PTC596 + PRIMA-1^MET^ + etoposide). The expression of the BMI-1 oncoprotein was also evaluated to quantify its levels under the described treatment conditions. As shown in Figure 4a, PTC596, given alone or in combination with PRIMA-1^MET^ and etoposide, reduced the BMI-1 levels of the HTLA-230 and HTLA-ER cell populations by approximately 35% and by 40% compared with the control, respectively. Notably, the exposure to PRIMA-1^MET^ or etoposide per se was not able to modify BMI-1 expression.

Furthermore, P53 protein expression was monitored under the tested treatment conditions, and as shown in Figure 4b,c, P53 levels did not change in either NB cell populations as well as the expression levels of p21, a P53-downstream protein, and MDM2, a negative modulator of P53 [59].

### 3.3. PTC596 Does Not Alter the Expression of Apoptotic Proteins in Parental and HTLA-ER Cells

In order to investigate whether the cytotoxic effects of the tested treatments were mediated by apoptosis induction, Bax and Bcl-2 protein levels were analysed. As shown in Figure 5a, the treatment of HTLA-230 cells with etoposide or PRIMA-1^MET^ increased the expression of the pro-apoptotic protein Bax by 36% and 24%, respectively, while it reduced the levels of anti-apoptotic Bcl-2 by 28% and 10%, respectively. Meanwhile, the same treatment conditions in HTLA-ER cells did not significantly modify Bax and Bcl-2 protein levels. In addition, the combined treatment (PTC596 + PRIMA1^MET^ + etoposide) in both NB cell populations was unable to alter Bax and Bcl-2 protein expressions (Figure 5a).

In fact, by calculating the Bax/Bcl-2 ratio, an increase of 80% and 35% was observed in HTLA-230 cells treated with etoposide and PRIMA-1^MET^, respectively, in comparison with untreated ones, confirming the induction of apoptosis in parental cells (Figure 5b).

### 3.4. PRIMA-1^MET^ and PTC596 Alone or in Combination Strongly Reduce the Clonogenic Potential of Parental and HTLA-ER Cells

The effects of PTC596, PRIMA1^MET^, and etoposide treatments on NB cell proliferation were assessed by using an anchorage-dependent clonogenicity assay. As shown in Figure 6a, all drugs, given alone or in combination, markedly inhibited the ability of HTLA-230 cells to form colonies. Similar results were observed in HTLA-ER cells, but, in this case, the treatment with etoposide alone did not significantly modify the clonogenic potential of control cells (Figure 6b).

### 3.5. PRIMA-1^MET^ and PTC596, Alone or in Combination, Reduce the Expression of Crucial EMT-Promoting Proteins and Inhibit CSC Formation

Since EMT is closely associated with cancer progression and resistance to therapy [60], the expressions of N-cadherin, ß-catenin, and SNAIL proteins were analysed. As shown in Figure 7a, etoposide per se decreased the expression of N-cadherin by 24% in HTLA-230 cells (left panel), while it did not modify this protein′s expression in HTLA-ER (right panel). Moreover, PRIMA-1^MET^ was able to reduce N-cadherin levels by 25% in both NB cell populations compared with untreated ones (Figure 7a). Meanwhile, PTC596 decreased N-cadherin expression by 60% in HTLA-230 and by 50% in HTLA-ER, and a similar trend was observed in the corresponding co-treated cells (Figure 7a).

As for ß-catenin, its expression level was reduced by 35% in etoposide-treated HTLA-230 cells, while it was not modified in etoposide-treated HTLA-ER cells (Figure 7b). PRIMA-1^MET^ was able to decrease ß-catenin by 45% in HTLA-230 cells and by 30% in HTLA-ER cells (Figure 7b), while PTC596 reduced ß-catenin by 70% and 40% in the parental and resistant cells, respectively (Figure 7b). Notably, the co-treatment with PTC596, PRIMA-1^MET^, and etoposide lowered ß-catenin by 70% in both NB cell populations in comparison with etoposide-treated ones and by 60% and 55% with respect to PRIMA-1^MET^-treated HTLA-230 and HTLA-ER cells, respectively (Figure 7b).

A similar modulation was observed for the expression of SNAIL proteins. In fact, PRIMA-1^MET^ reduced SNAIL expression by approximately 35% in both NB cells, while PTC596 reduced SNAIL expression by 46% and 70% in HTLA-230 and HTLA-ER cells, respectively (Figure 7c). Indeed, the co-treatment with PTC596, PRIMA-1^MET^, and etoposide decreased SNAIL levels by 70% in both NB cells in comparison with etoposide-treated ones (Figure 7c).

Considering that EMT has been well documented to have a strong overlap with the development of the CSC phenotype [61,62,63,64] and that EMT and CSCs have been demonstrated to be determining factors for the onset of chemoresistance [48,65], the ability to generate CSCs was also evaluated. As shown in Figure 8, PRIMA-1^MET^, PTC596 alone, and their combination with etoposide were able to inhibit CSC formation in HTLA-230 and HTLA-ER cells.

### 3.6. PRIMA-1^MET^ and PTC596, Alone or in Combination, Enhance H_2_O_2_ Production, Deplete Intracellular GSH, and Induce Lipid Peroxidation

Considering that the cytotoxic effect of etoposide can be mediated by ROS production [66], the amount of H_2_O_2_, the most long-lived ROS [67,68], was monitored in both NB cell lines under each treatment condition. As shown in Figure 9a, etoposide exposure increased H_2_O_2_ levels in HTLA-230 by 10%, whereas, consistent with our previous study [43], no significant changes were observed in HTLA-ER. Meanwhile, PRIMA-1^MET^ per se enhanced H_2_O_2_ production by 20% only in parental cells (Figure 9a).

Interestingly, PTC596 alone increased H_2_O_2_ levels by 35% and 40% in HTLA-230 and HTLA-ER, respectively (Figure 9a), while the combined co-treatment (PTC596, PRIMA-1^MET^, and etoposide) induced a 120% and 55% increase in H_2_O_2_ production in HTLA-230 and HTLA-ER cells in comparison with the untreated ones, respectively (Figure 9a). Notably, under the co-treatment conditions, H_2_O_2_ levels increased by 100% and 67% in HTLA-230 and HTLA-ER cells in comparison with etoposide, respectively; by 80% and 65% in comparison with PRIMA-1^MET^, respectively; and by only 60% in HTLA-230 with respect to PTC596 (Figure 9a).

Since GSH plays a crucial role in cancer progression and chemoresistance [8,43,69], GSH levels were monitored under each treatment condition and the total GSH content in untreated HTLA-ER cells was found to be double (0.418 µM/µg pt) in comparison with HTLA-230 cells (0.209 µM/µg pt). In agreement with our previous results [43], etoposide per se did not alter GSH levels in both NB cell populations (Figure 9b), while PRIMA-1^MET^ and PTC596 were able to reduce GSH by 25% and 32% in HTLA-230, respectively, and by 20% and 60% in HTLA-ER, respectively, compared with untreated cells (Figure 9b). Moreover, a further reduction in GSH content was observed in the co-treated cells. In detail, a 60% and 70% decrease in GSH was found in the co-treated HTLA-230 and HTLA-ER cells respectively, in comparison with the corresponding etoposide-treated ones (Figure 9b). The amount of GSSG, the oxidised form of GSH, in etoposide-treated cells was similar to that in untreated cells, whereas it was below the detection limit under each other treatment condition.

Interestingly, HPLC analysis also showed that cystine and cysteine levels, in untreated HTLA-ER cells were 65% and 70% higher than those detected in HTLA-230 cells, respectively, in agreement with the fact that resistant cells, having more GSH, need more of the precursor.

Lastly, the involvement of membrane lipid oxidative degradation in the induction of cell damage [70] was evaluated by analysing lipid peroxidation (LPO). As shown in Figure 9c, PRIMA-1^MET^ alone increased LPO by 30% only in HTLA-230 cells, while PTC596 was able to enhance LPO by 40% in both NB cell populations. Meanwhile, the combined treatment increased LPO by 70% and 43% in HTLA-230 and HTLA-E, in comparison with etoposide, respectively; by 40% in both cell populations with respect to PRIMA-1^MET^; and by 25% only in parental cells with respect to PTC596 (Figure 9c).

### 3.7. PTC596 Exerts Its Cytotoxic Effect by Inducing Ferroptosis

Based on the evidence that PTC596 alone and in combination with etoposide and PRIMA-1^MET^ was more effective in inducing H_2_O_2_ overproduction, GSH depletion, lipid peroxidation, and non-apoptotic cell death, the hypothesis of the induction of ferroptosis, a newly described type of programmed cell death, was considered. Therefore, HTLA-230 and HTLA-ER were treated with ferrostatin or liprostatin, as ferroptosis-inhibitory agents, and with RSL3 or erastin, as ferroptosis-inducing compounds, for 48 h and 72 h, either given alone or co-administered with PTC596.

As shown in Figure 10a, ferrostatin or liprostatin, given alone, did not alter the viability of both cell populations, while their co-treatment with PTC596 strongly limited its cytotoxic effect, thus supporting the hypothesis that PTC596 is able to induce ferroptosis.

On the other hand, RSL3 alone reduced the viability of HTLA-230 cells by 43% (after 48 h) and 47% (after 72 h) and of HTLA-ER cells by 30% (Figure 10b), while the co-treatment with PTC596 was able to further reduce the viability of both NB cell lines by 15% (Figure 10b) in comparison with the untreated ones. Moreover, 48 h and 72 h treatments with the other ferroptosis inducer erastin decreased the viability of parental and resistant cells by 50% and 20%, respectively, and the co-treatment with PTC596 led to a further 13% reduction (Figure 10b).

In order to better investigate the pathway involved in ferroptosis induction, the activity of GPX4, the first line of cell defence against ferroptosis [71], was evaluated. In this context, the basal GPX4 activity in HTLA-ER cells was 60% higher (1.55 ± 0.005 nmol/min/mg) than that found in HTLA-230 cells (0.97 ± 0.01 nmol/min/mg). Meanwhile, the treatments with PTC596 for 48 and 72 h were not able to significantly modify the basal GPX4 activity both in parental and resistant cells.

These results demonstrate, on the one hand, that GPX4 activity is potentially involved in NB drug resistance and, on the other, that ferroptosis induced by PTC596 is not dependent on GPX4.

## 4. Discussion

Long-term therapy induces several genetic and epigenetic changes that are responsible for cancer cells′ adaptive response to a broad spectrum of structurally unrelated compounds, leading to the onset of multidrug resistance (MDR) [5,72,73,74]. In this context, our findings show that chronic exposure to etoposide is able to select a population of NB cells (HTLA-ER) resistant not only to etoposide but also to the anticancer drugs included in the induction therapy commonly used to treat NB patients (i.e., carboplatin, cisplatin, doxorubicin, cyclophosphamide, and vincristine). According to a wide range of literature, we have previously shown that drug-resistant cancer cells are able to maintain efficient mitochondrial respiration and activate GSH-related antioxidant responses [43,75,76,77,78,79]. Moreover, although HTLA-ER cells show the same homozygous *TP53* missense mutation (p.A161T) of parental cells, they display a monoallelic deletion of the *13q14.3* locus with a consequent downregulation of the miRNA 15a/16-1 family and BMI-1 upregulation [28].

Therefore, based on these findings, we hypothesised that PTC596, an inhibitor of BMI-1, a protein able to downmodulate intracellular GSH levels [21], and PRIMA-1^MET^, a compound capable of reactivating mutant P53 and binding GSH [80], might represent a potential strategy to fight chemoresistance.

Herein, we reported that PTC596 (i) reduces BMI-1 protein levels in both NB cell populations, although HTLA-ER cells exhibit a higher basal expression of BMI-1 in comparison with the parental cells [28], and (ii) decreases their cell viability without inducing apoptosis.

As for PRIMA-1^MET^, we showed that (i) it does not alter either the expression levels of p53 or those of MDM2 and p21 in both NB cell populations, and (ii) it is able to trigger the apoptosis of parental cells only. This suggests that although both cell populations have the same TP53 mutation, this mutation is not linked to the onset of MDR [28], and we suppose that the resistant cells must have other mechanisms to evade apoptotic cell death.

In addition, while the co-administration of PTC596, PRIMA-1^MET^, and etoposide has a synergistic cytotoxic effect on HTLA-230, PTC596 is more cytotoxic to HTLA-ER than PRIMA-1^MET^ and etoposide. These results support the role of BMI-1 in the onset of chemoresistance according to several clinical data [32,81,82]. In this context, it has been found that BMI-1 inhibition increases the sensitivity of head and neck CSCs to cisplatin [37] and that cisplatin resistance is due to the ability of BMI-1 to promote cancer cell growth and proliferation by activating the PI3K/Akt-dependent pathway [83].

Furthermore, in our study, the analysis of the clonogenic potential shows that PRIMA-1^MET^ is effective in reducing the clonogenicity of both NB cell populations, in agreement with a study reporting the ability of PRIMA-1^MET^ to counteract glioblastoma growth by reducing colony formation [84]. Similarly, PTC596 is effective at strongly inhibiting the clonogenic potential of both NB cell populations. These findings are in line with previous studies demonstrating that BMI-1 inhibition downregulates the clonogenic potential of myeloma cells [85] and oesophageal cancer cells [35].

To further validate the hypothesis of a possible therapeutic approach, based on targeting P53 and BMI-1, for counteracting cancer growth and MDR, the effects of PRIMA-1^MET^ and PTC596 on EMT were evaluated. Notably, EMT is a crucial step in the acquisition of the hallmarks of aggressive cancer such as cell plasticity, metastasis formation, the CSC phenotype, and therapy resistance. In this context, we showed that PRIMA-1^MET^ and PTC596 were able to markedly reduce the expression of EMT proteins in both NB cell populations. Moreover, this effect was further amplified via the co-treatments of PTC596 with etoposide and PRIMA-1^MET^, suggesting a synergistic action of these compounds in downregulating EMT. With regard to PRIMA-1^MET^, this result might be supported by the fact that P53 can transcriptionally regulate EMT genes and that EMT also depends on P53 status [86]. Moreover, specific mutations in the *TP53* gene can convert the onco-suppressor into an oncogene, leading to EMT-related phenotypic alterations [86].

Analogously, PTC596 has been demonstrated to counteract EMT, in agreement with the literature reporting that BMI-1, the target of PTC596, promotes cancer migration and invasion by modulating the expression of EMT-related proteins [30].

Moreover, considering that EMT is strictly associated with the acquisition of the CSC phenotype [61,62,63,64], crucially involved in the onset of chemoresistance [48,63], it is not surprising that PRIMA-1^MET^ and PTC596 are effective in inhibiting CSC generation in HTLA-230 and, more importantly, in HTLA-ER cells. With regard to the effect of PRIMA-1^MET^, our results are in line with the study by Patyka M et al., reporting that the drug is effective in preventing the propagation of neurospheres originating from patient-derived glioblastoma cells [84]. Similarly, this ability has been also recently described for PTC596 [87,88], suggesting that the modulation of BMI-1 might be a promising approach to curing cancer patients.

In addition, our findings show that PRIMA-1^MET^ and PTC596 induce a decrease in GSH levels and ROS overproduction, supporting the hypothesis that their ability to alter the cellular redox balance can be used to trigger cancer cell death and fight drug resistance.

The results concerning PRIMA-1^MET^′s cytotoxic effect based on GSH depletion and oxidative stress induction are in line with several studies [18,89,90,91,92,93,94]. This is the first time that PTC596 has been demonstrated to decrease GSH levels in MDR cells. Although further investigations are needed to understand the molecular mechanisms whereby PTC596 can modulate intracellular GSH content, it is reasonable to believe that this response is the consequence of the inhibition of BMI-1, which has been demonstrated to regulate GSH production [21,43].

Collectively, our findings suggest that PTC596, given alone or in combination with PRIMA-1^MET^ and etoposide, by reducing the intracellular GSH amount, induces the oxidative death of both parental and MDR NB cells.

Lastly, the induction of lipid peroxidation allows us to hypothesize that PTC596 exerts its cytotoxic action by inducing ferroptosis, a kind of programmed cell death triggered by an excessive production of ROS and resulting in the peroxidation of membrane phospholipids [71,95,96,97]. The results obtained by co-treating both NB cell populations with PTC596 and ferroptosis inhibitors (liprostatin and ferrostatin) or activators (RSL3 and erastin) demonstrate, for the first time, that PTC596 is able to induce ferroptotic death.

With regard to the potential pathway involved, our data suggest that PTC-induced ferroptosis is not mediated by GPX4 activity, leading to the hypothesis that other mechanisms might be involved [98].

MYCN-amplified NBs are particularly sensitive to ferroptosis since MYCN upregulates iron influx and activates the Xc system, a cysteine transporter, creating a marked dependence on GSH-related pathways [99,100]. In addition, MYCN overexpression has been demonstrated to upregulate the expression of the TFRC gene, which encodes transferrin receptor 1, an iron transporter protein on the cell membrane [101].

Regarding this connection, future studies aimed at investigating the potential relationship between ferroptotic PTC action and the MYCN state could be important to overcome MDR in NB.

Therefore, the results obtained via this in vitro model of MDR could be a good starting point for the validation in clinical research in order to identify new potential strategies that are able to enhance the sensitivity of NB to anticancer therapy.

## 5. Conclusions

Taken together, our results suggest that as long as MDR cells are able to maintain redox homeostasis, the overexpression of the oncoprotein BMI-1, and high levels of GSH, drug refractoriness is maintained. Therefore, an approach that can simultaneously regulate these molecular targets and induce ferroptosis, as proposed with PTC596, could be the winning strategy to counteract chemoresistance. Importantly, the activation of ferroptosis could be exploited to more selectively target NB cells and especially MDR cells, which are characterised by higher amounts of GSH compared with healthy cells.

## Figures and Tables

**Figure 1 antioxidants-13-00003-f001:**
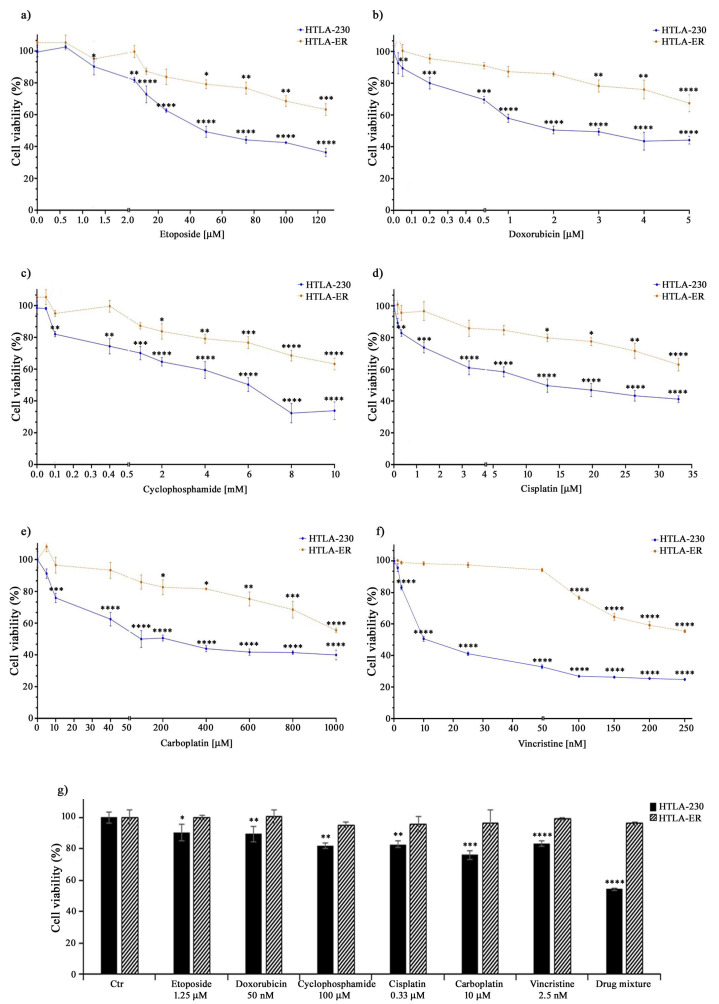
HTLA-ER cells acquired multidrug resistance. Cell viability was evaluated using MTS assay in HTLA-230 (blue lines) and in HTLA-ER cells (orange lines) treated with 0.625–125 µM etoposide (**a**), 0.025–5 µM doxorubicin (**b**), 0.05–10 mM cyclophosphamide (**c**), 0.165–33 µM cisplatin (**d**), 5–1000 µM carboplatin (**e**), and 1.25–250 nM vincristine (**f**) for 48 h. (**g**) reports the percentage viability of cells treated for 48 h with the clinically used doses of the compounds given alone or in a drug mixture (as in the induction therapy; last pair of columns) in comparison with that of untreated ones (100%). Histograms summarize quantitative data of means ± SEM of at least four independent experiments. * *p* < 0.1; ** *p* < 0.01; *** *p* < 0.001; and **** *p* < 0.0001 vs. untreated HTLA-230 or HTLA-ER cells.

**Figure 2 antioxidants-13-00003-f002:**
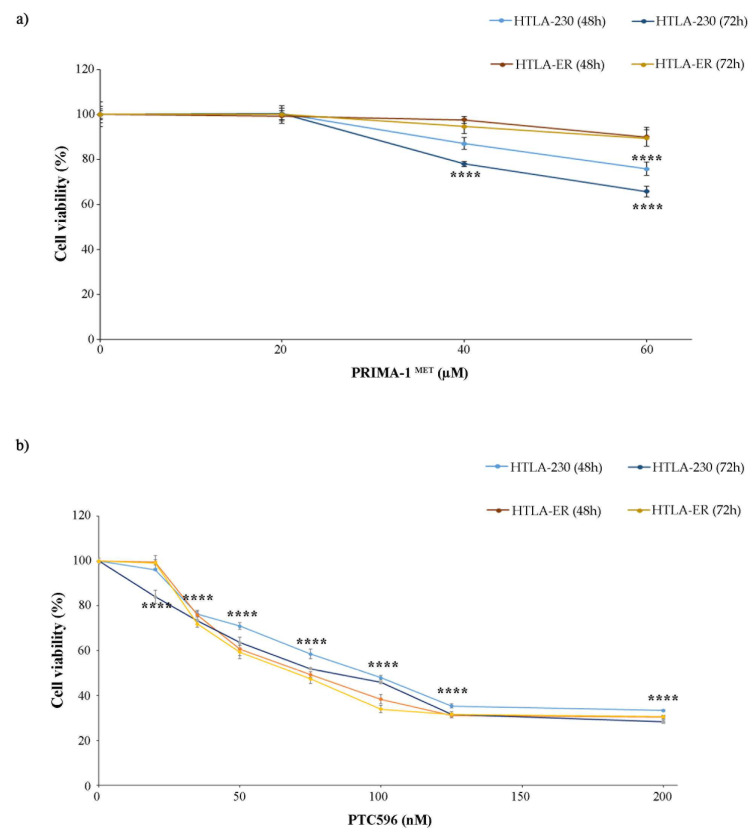
Selection of PRIMA-1^MET^ and PTC596 dose treatments. Cell viability was evaluated using MTS assay in cells exposed to increasing concentrations of the drugs. (**a**) HTLA-230 and HTLA-ER cells were treated with 10–60 µM of PRIMA-1^MET^ for 48 h and 72 h. (**b**) HTLA-ER cells were treated with 20–200 nM of PTC596 for 48 h and 72 h. Results are expressed as percentage variations in cell viability of treated cells with respect to that of untreated ones (100%). Graphs summarize quantitative data of means ± SEM of at least four independent experiments. **** *p* < 0.0001 vs. untreated HTLA-230 or HTLA-ER cells.

**Figure 3 antioxidants-13-00003-f003:**
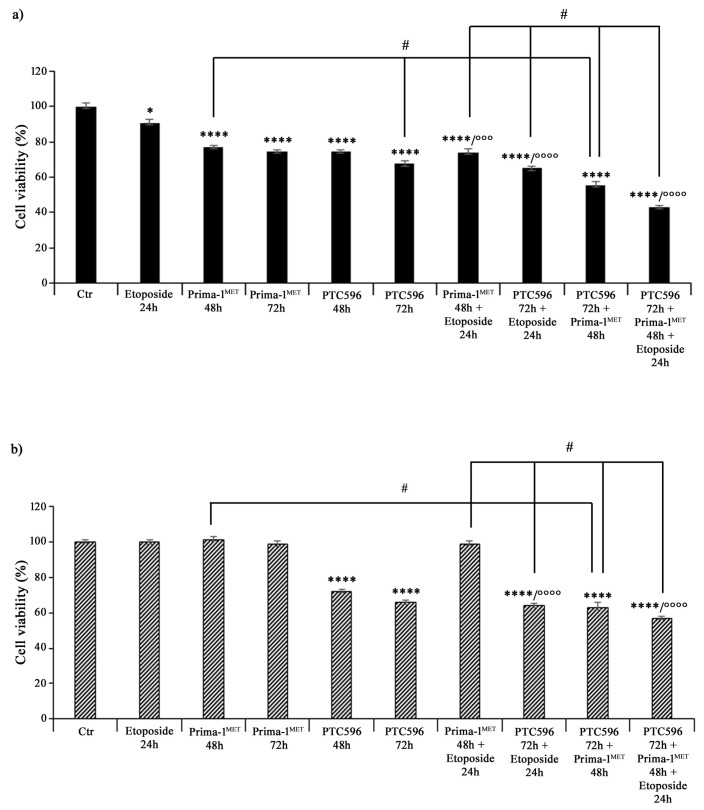
PRIMA-1^MET^ and PTC596, used in combination, were cytotoxic to HTLA-230 and HTLA-ER cells. Cell viability was evaluated using MTS assay in HTLA-230 (**a**) and HTLA-ER (**b**) cells treated with 1.25 μM of etoposide (24 h), 60 μM of PRIMA-1^MET^ (48 h and 72 h), and 35 nM of PTC596 (48 h and 72 h), given alone or in each possible combination. Results are expressed as percentage variations in cell viability of treated cells with respect to that of untreated ones (100%). Histograms summarize quantitative data of means ± SEM of at least four independent experiments. * *p* < 0.1; **** *p* < 0.0001 vs. untreated HTLA-230 or HTLA-ER cells (Ctr); °°° *p* < 0.001; °°°° *p* < 0.0001 vs. etoposide treated cells; # *p* < 0.0001 vs. respective treatments indicated by the bars.

**Figure 4 antioxidants-13-00003-f004:**
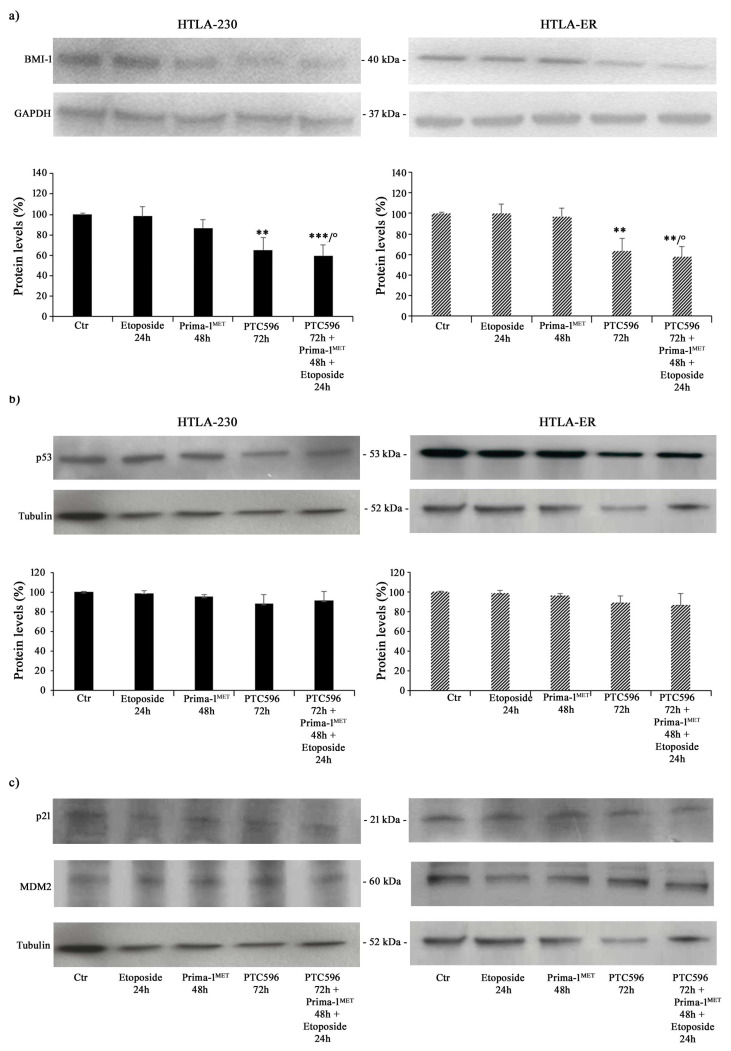
PTC596, alone or in combination, reduced the expression levels of BMI-1, while PRIMA-1^MET^ did not alter the expression of P53 and its related proteins. Protein levels of BMI-1 (**a**), P53 (**b**), p21, and MDM2 (**c**) in HTLA-230 (left panel) and HTLA-ER (right panel) cells untreated and treated with 1.25 μM of etoposide (24 h), 60 μM of PRIMA-1^MET^ (48 h), and 35 nM of PTC596 (72 h), given alone or in combination. Immunoblots shown are representative of four independent experiments. GAPDH (**a**) or tubulin (**b**,**c**) were the internal loading controls used to normalize the protein levels of BMI-1 and P53, respectively. Results are expressed as percentage variations in protein levels in treated cells with respect to that in untreated ones (100%). Histograms summarize quantitative data of protein level means, normalised to the respective loading control ± SEM of four independent experiments. ** *p* < 0.01; *** *p* < 0.001 vs. untreated cells (Ctr); ° *p* < 0.0001 vs. etoposide-treated cells.

**Figure 5 antioxidants-13-00003-f005:**
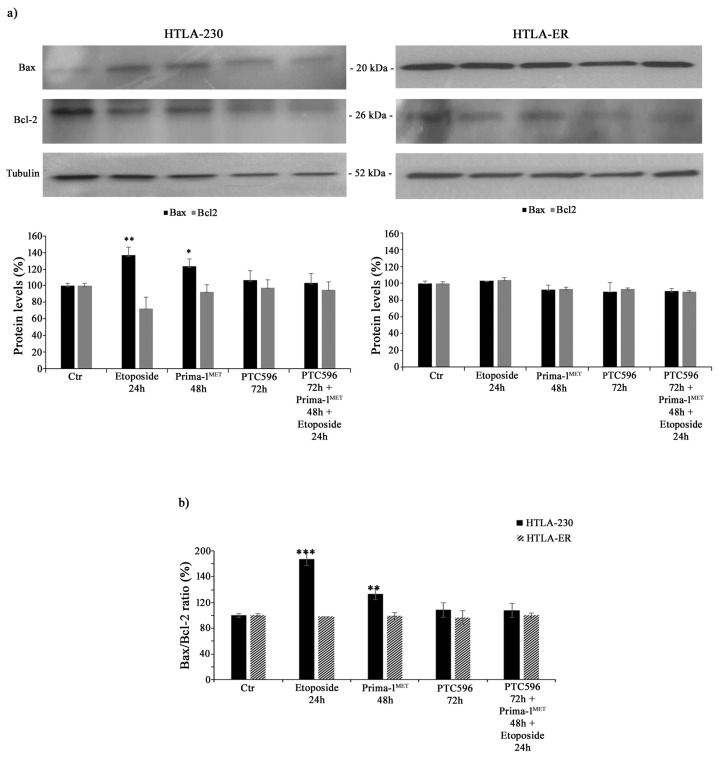
PTC596, alone or in combination, does not induce apoptosis. Protein levels of Bax and Bcl-2 (**a**) in HTLA-230 (left panel) and HTLA-ER (right panel) cells untreated or treated with 1.25 μM of etoposide (24 h), 60 μM of PRIMA-1^MET^ (48 h), and 35 nm of PTC596 (72 h), given alone or in combination. The histogram reported in (**b**) summarizes the values of Bax/Bcl-2 ratio. Immunoblots shown are representative of four independent experiments. Tubulin is the internal loading control. Results are expressed as percentage variations in protein levels in treated cells with respect to that in untreated ones (100%). Histograms summarize quantitative data of protein level means, normalised to tubulin ± SEM of four independent experiments. * *p* < 0.1; ** *p* < 0.01; *** *p* < 0.001 vs. untreated cells (Ctr).

**Figure 6 antioxidants-13-00003-f006:**
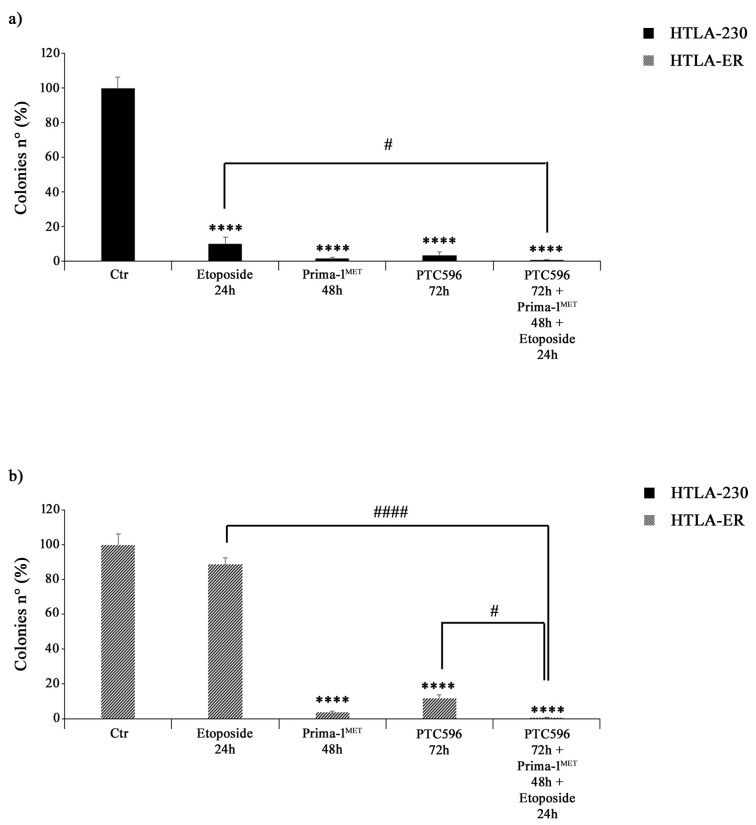
PRIMA-1^MET^ and PTC596, alone or in combination, inhibit the clonogenic potential of HTLA-230 and HTLA-ER cells. The ability to form colonies was analysed by using an anchorage-independent clonogenic assay. HTLA-230 (**a**) and HTLA-ER (**b**) cells were treated with 1.25 μM of etoposide (24 h), 60 μM of PRIMA-1^MET^ (48 h), and 35 nM of PTC596 (72 h), given alone or in combination. Results are expressed as percentage variations in colony number of treated cells with respect to that of untreated ones (100%). The histograms summarize quantitative data of means ± SEM of four independent experiments. **** *p* < 0.0001 vs. untreated cells (Ctr); # *p* < 0.1; #### *p* < 0.0001 vs. respective treatments indicated by the bars.

**Figure 7 antioxidants-13-00003-f007:**
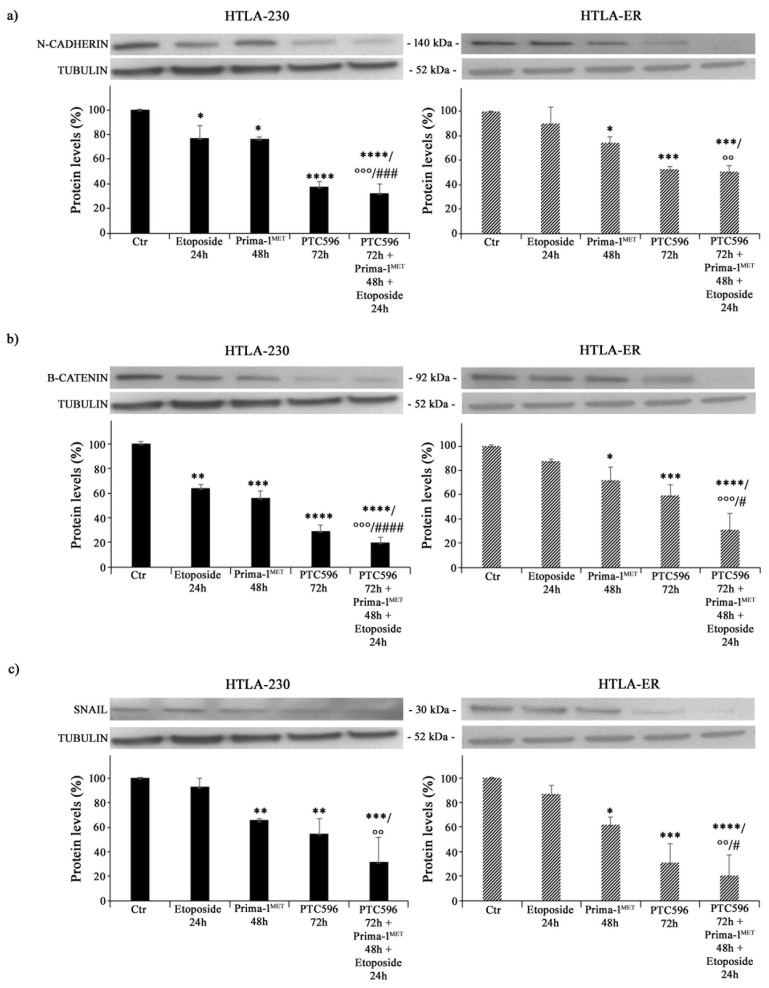
PRIMA-1^MET^ and PTC596, alone or in combination, inhibit the expression of EMT-related proteins. Protein levels of N-cadherin (**a**), ß-catenin (**b**), and SNAIL (**c**) in HTLA-230 (left panels) and HTLA-ER (right panels) cells untreated or treated with 1.25 μM of etoposide (24 h), 60 μM of PRIMA-1^MET^ (48 h), and 35 nM of PTC596 (72 h), given alone or in combination. Immunoblots shown are representative of four independent experiments. Tubulin is the internal loading control. Results are expressed as percentage variations in protein levels in treated cells with respect to that in untreated ones (100%). The histograms summarize quantitative data of protein level means, normalised to tubulin ± SEM of four independent experiments. * *p* < 0.1; ** *p* < 0.01; *** *p* < 0.001; **** *p* < 0.0001 vs. untreated cells (Ctr); °° *p* < 0.01; °°° *p* < 0.001 vs. etoposide-treated cells; # *p* < 0.1; ### *p* < 0.001; #### *p* < 0.0001 vs. PRIMA-1^MET^-treated cells.

**Figure 8 antioxidants-13-00003-f008:**
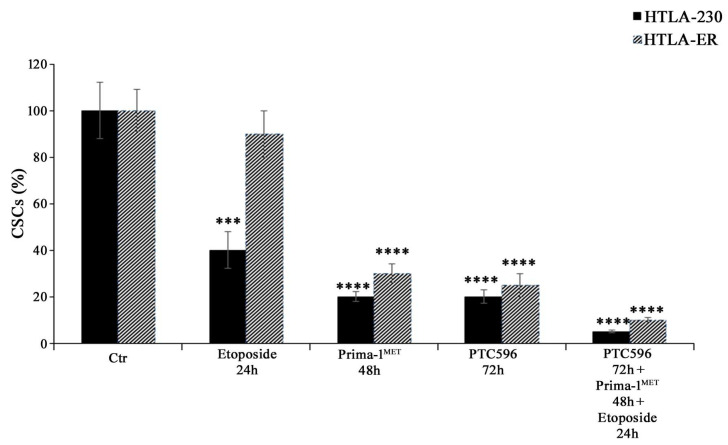
PRIMA-1^MET^ and PTC596, alone or in combination, totally counteract CSC generation. Evaluation of CSC number in HTLA-230 and HTLA-ER cells untreated or treated with 1.25 μM of etoposide (24 h), 60 μM of PRIMA-1^MET^ (48 h), and 35 nM of PTC596 (72 h), given alone or in combination. The number of cells obtained by the disaggregation of CSCs, generated after 4 weeks, is reported as percentage values of the number of treated cells in comparison with those of untreated ones (100%). The histograms summarize quantitative data of means ± SEM of four independent experiments, 4 weeks after the treatments. *** *p* < 0.001; **** *p* < 0.0001 vs. untreated cells (Ctr).

**Figure 9 antioxidants-13-00003-f009:**
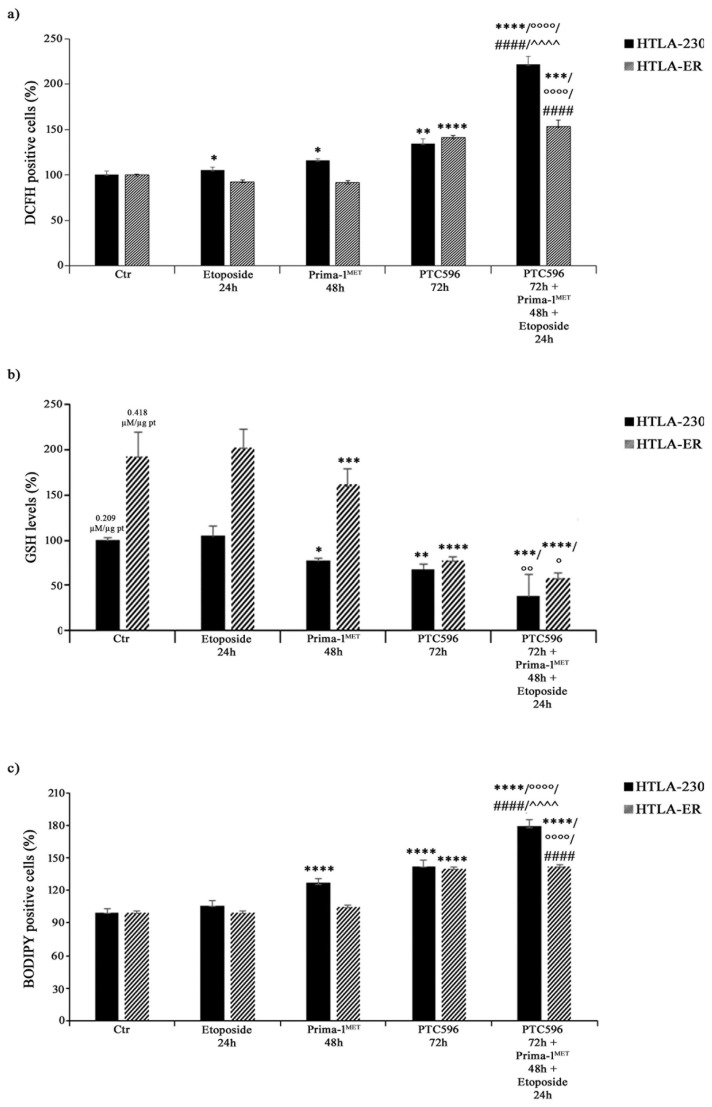
PRIMA-1^MET^ and PTC596, alone or in combination, enhance H_2_O_2_ production, reduce GSH intracellular levels, and induce lipoperoxidation. H_2_O_2_ production (**a**), GSH levels (**b**), and lipid peroxidation (BODIPY-positive cells (**c**) were analysed in HTLA-230 and HTLA-ER cells untreated or treated with 1.25 μM of etoposide (24 h), 60 μM of PRIMA-1^MET^ (48 h), and 35 nM of PTC596 (72 h), given alone or in combination). Results are expressed as percentage variations in DCFH-positive cells, GSH levels, or BODIPY-positive cells under treatment conditions in comparison with untreated ones (100%). Histograms summarise quantitative data of means ± SEM of four independent experiments. * *p* < 0.1; ** *p* < 0.01; *** *p* < 0.001; **** *p* < 0.0001 vs. untreated cells (Ctr); ° *p* < 0.1; °° *p* < 0.01; °°°° *p* < 0.0001 vs. etoposide-treated cells; #### *p* < 0.0001 vs. PRIMA-1^MET^-treated cells; ^^^^ *p* < 0.0001 vs. PTC596-treated cells.

**Figure 10 antioxidants-13-00003-f010:**
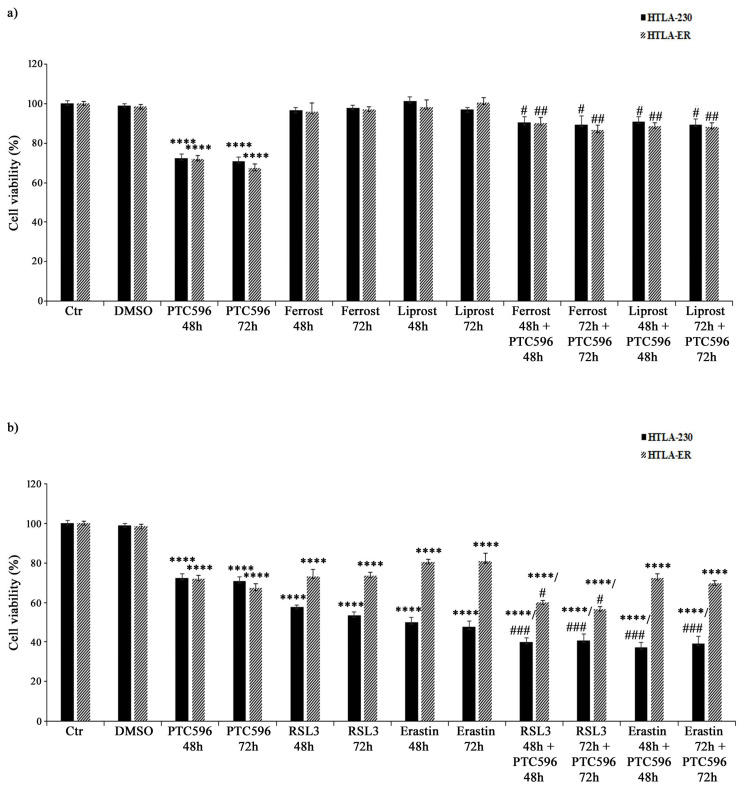
PTC596 induces ferroptosis in parental and HTLA-ER cells. Cell viability was evaluated using MTS assay in HTLA-230 and HTLA-ER cells treated for 48 h or 72 h with 35 nM of PTC596 and ferroptosis-inhibitory drugs ((**a**) 5 μM of ferrostatin and 1 μM of liprostatin) or ferroptosis-inducing compounds ((**b**) 250 nM of RSL3 and 2.5 μM of erastin), given alone or in combination. Results are expressed as percentage variations in the viability of treated cells with respect to that of untreated ones (100%). Histograms summarize quantitative data of means ± SEM of four independent experiments. **** *p* < 0.0001 vs. untreated cells (Ctr); # *p* < 0.1 vs. PTC596-treated cells; ## *p* < 0.01 vs. PTC596-treated cells; ### *p* < 0.001 vs. PTC596-treated cells.

**Table 1 antioxidants-13-00003-t001:** IC_50_ evaluated in HTLA-230 and HTLA-ER cells exposed to increasing concentrations of the chemotherapeutic drugs tested for 48 h.

Drug	Dose Compared with That Clinically Used	HTLA-230 Cells’ IC_50_	HTLA-ER Cells’ IC_50_
Etoposide (μM)	1.25	69.79	221.4
Doxorubicin (μM)	0.05	3.21	8.88
Cyclophosphamide (mM)	0.1	5.30	14.86
Cisplatin (μM)	0.33	19.78	54.58
Carboplatin (μM)	10	509.3	1345.02
Vincristine (nM)	2.5	51.6	234.66

## Data Availability

The data presented in this study are available on request from the corresponding author.

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
