# Peer review of "PTC596-Induced BMI-1 Inhibition Fights Neuroblastoma Multidrug Resistance by Inducing Ferroptosis"

_antioxidants, 2023, doi:10.3390/antiox13010003_

Round 1

Reviewer 1 Report

Comments and Suggestions for Authors

Author Response

Dear Editor,

Firstly, we would like to thank you for having reviewed the paper and thank the Reviewers for their most welcome and helpful comments that have gone towards improving the quality of the manuscript which is now enclosed.

In this revision phase, we have tried as far as possible to satisfy the requests and comments by modifying the main text. For greater clarity, the changes made to the original manuscript were made by using the Track Changes system (Word).

As you will see, we have addressed all the points raised by the Reviewers and amended the manuscript accordingly as follows:

REVIEWER 1

Valenti and coworkers developed an interesting study concerning the inhibition of BMI-1 in two neuroblastoma cell lines, HTLA-230 and HTLA-ER, both showing N-Myc amplification and p53 mutation. In addition, HTLA-ER developed a MDR phenotype by exposure to Etoposide. The rationale of the study is clear, the experiments coherent with the aims and well conducted. The finding is very interesting, since, so far, no study has linked the inhibition of BMI-1 and ferroptosis in neuroblastoma or other human cancer. As ferroptosis might be triggered when apoptosis is not working, BMI-1 inhibition might represent an opportunity for intervention in very aggressive and drug-resistant cancer.

We thank the Reviewer for the positive comments and appreciation.

General comment: The very interesting finding is that BMI-1 inhibition activates ferroptosis. Ferroptosis can be related with the intracellular abundance of iron, of which cancer cells are particularly avid. Several studies indicate a relation between N-Myc amplification (and/or overexpression) and iron avidity that might induce ferroptosis as a response to oxidative stress. This point ought to be discussed too, since the presence of iron might condition the response to therapy, switching the cell death program from apoptosis (often not working) to ferroptosis.

We absolutely agree with the Reviewer that a relationship between N-Myc amplification, iron and ferroptosis exist and we have discussed this point at lines 627-634 of the revised version.

Possibly, a second neuroblastoma cell line having N-Myc amplification might be subject of a few experiments to validate whether, beyond drug resistance, N-Myc amplification and iron content can induce ferroptosis by BMI-1 inhibition. This might greatly support the paradigm that targeting BMI-1 in neuroblastoma offers the opportunity of a new therapeutic approach.

We agree with the Reviewer but, considering that the selection of a second drug resistant cell population would take about 6 months of experimental work, this suggestion could be a matter for a future study aimed at confirming the paradigm that targeting BMI-1 in neuroblastoma offers the opportunity of a new therapeutic approach.  

In the present study, the pathway leading to ferroptosis is not defined. The study seems a bit “unbalanced”: the numerous experiments on the different action of PTC596 and PRIMA-1 MET are not completed by enough experiments to support how ferroptosis is triggered. Data concerning the ferroptosis mechanism of action (which genes are up-regulated and which pathway is involved) are mandatory to strengthen the findings. I agree that invasion, CSC forming, EMT and clonogenic assay are fundamental to define the activity of PTC596 and PRIMA-1 MET , but these aspects are less intriguing than ferroptosis: this experimental part ought to be increased.

We agree with the Reviewer that it is important to clarify how ferroptosis is triggered in our model. To this end, the activity of GPX4, the first line of cell defence against ferroptosis, has now been evaluated. The results have been reported at the end of paragraph 3.7 (lines 534-538) and discussed at lines 624-626 of the Discussion section.

Minor comments:

Fig.5a, the WB image is not very clear. (?)

In agreement with the Reviewer, the graph reporting Bax/Bcl2 ratio has now been modified in order to make Fig. 5a more explanatory.

Discussion, line 545-548. Although p53 quantity was not found to change, p53 function could have changed: a simple WB cannot state it. Downstream gene and/or protein expression could better support this statement. As p53 activity can also regulate ferroptosis in both directions (activation/inhibition) the experiment reported in the paragraph 3.7. PTC596 exerts its cytotoxic effect by inducing ferroptosis ought to be performed with PRIMA-1 MET too and with the 2 drugs together.

As suggested by the Reviewer, the expression of p21 and MDM2, two p53-related proteins, have been evaluated (Figure 4) to better investigate the role of p53.

Ferroptosis inducing/inhibitors have been tested only in the presence of PTC596 treatment because the changes in the parameters related to ferroptosis induction (H2O2 overproduction, GSH depletion and lipoperoxidation) detected in resistant cells treated with PTC were the same as those from the resistant cells co-treated with PRIMA-1 MET.

Reviewer 2 Report

Comments and Suggestions for Authors

Manuscript ID: antioxidants-2727049

PTC596-induced BMI-1 inhibition fights neuroblastoma multidrug resistance by inducing ferroptosis

Authors Giulia Elda Valenti, Antonella Roveri , Rina Venerando , Paola Menichini , Paola Monti , Bruno Tasso, Nicola Traverso , Cinzia Domenicotti, Barbara Marengo

Authors explored the activity PTC596, an inhibitor of BMI-1, alone or in combination with other anticancer agents with the aim of evaluating its potential as a therapeutic agent able to counteract multidrug resistance. The experimental study was entirely performed in in vitro model of neuroblastoma and various cell processes were analyzed. Eventually, they found out that the mechanism of action involved the induction of oxidative stress and ferroptosis.

The study identified PTC596 as a highly active compound which could be accordingly introduced as an anticancer agent with low chemoresistance in neuroblastoma therapy. The paper is clearly written and is suitable to be published in Antioxidants. However, I have minor comments to be answered before acceptance for publication:

1)      Figure 5b, the graph should be labeled as % in the y-axis and SEM bars in the plot.

2)      Authors should provide the entire blotting membranes.

3)      The study was performed only in in vitro models: in discussion the limitation of an in vitro model should be argued, especially considering the translational potential.

Author Response

Dear Editor,

Firstly, we would like to thank you for having reviewed the paper and thank the Reviewers for their most welcome and helpful comments that have gone towards improving the quality of the manuscript which is now enclosed.

In this revision phase, we have tried as far as possible to satisfy the requests and comments by modifying the main text. For greater clarity, the changes made to the original manuscript were made by using the Track Changes system (Word).

As you will see, we have addressed all the points raised by the Reviewers and amended the manuscript accordingly as follows:

REVIEWER 2

Authors explored the activity PTC596, an inhibitor of BMI-1, alone or in combination with other anticancer agents with the aim of evaluating its potential as a therapeutic agent able to counteract multidrug resistance. The experimental study was entirely performed in in vitro model of neuroblastoma and various cell processes were analyzed. Eventually, they found out that the mechanism of action involved the induction of oxidative stress and ferroptosis. The study identified PTC596 as a highly active compound which could be accordingly introduced as an anticancer agent with low chemoresistance in neuroblastoma therapy.The paper is clearly written and is suitable to be published in Antioxidants.

We thank the Reviewer for the positive comments and appreciation.

However, I have minor comments to be answered before acceptance for publication:

  • Figure 5b, the graph should be labeled as % in the y-axis and SEM bars in the plot.

As required by the Reviewer, the graph has been labeled as % and SEM bars have been added.

  • Authors should provide the entire blotting membranes.

The non cropped blotting membranes have been already reported in the “Supporting information file” submitted together with the paper following the Author’s guidelines.

  • The study was performed only in in vitro models: in discussion the limitation of an in vitro model should be argued, especially considering the translational potential.

We agree with the Reviewer comment and, as suggested, the limitation of the in vitro model has been argued at lines 635-637.